# MRSA Femoral Osteomyelitis from Superinfected Scabies Lesions: A Pediatric Case Report

**DOI:** 10.3390/ijerph19021007

**Published:** 2022-01-17

**Authors:** Marco Ugo Andrea Sartorio, Alice Marianna Munari, Patrizia Carlucci, Paola Erba, Valeria Calcaterra, Valentina Fabiano

**Affiliations:** 1Department of Pediatrics, Vittore Buzzi Children’s Hospital, University of Milan, Via Castelvetro 32, 20154 Milan, Italy; patrizia.carlucci@asst-fbf-sacco.it (P.C.); paola.erba@asst-fbf-sacco.it (P.E.); valeria.calcaterra@asst-fbf-sacco.it (V.C.); valentina.fabiano@unimi.it (V.F.); 2Pediatric Radiology and Neuroradiology, Vittore Buzzi Children’s Hospital, 20154 Milan, Italy; alice.munari@asst-fbf-sacco.it; 3Pediatrics and Adolescentology Unit, Department of Internal Medicine and Therapeutics, University of Pavia, 27100 Pavia, Italy

**Keywords:** ectoparasitic infestations, methicillin-resistant *Staphylococcus aureus*, neglected disease, osteomyelitis, pediatrics, public health, scabies, superinfection

## Abstract

Scabies is a skin infestation from the *Sarcoptes scabiei*. It is considered a public health issue causing concern in developing countries and is considered a “neglected tropical disease” by the World Health Organization (WHO). Scabies skin lesions may cause severe itching and can be the portal of entry for opportunistic and pathogenic bacteria, which can cause serious systemic infections. We report the case of a 3-year-old boy with recurrent scabies infections who presented to the emergency department because of a fever and refusal to walk. Blood tests showed neutrophilic leukocytosis and significantly increased C reactive protein (CRP) and procalcitonin. Upon medical examination, his right thigh was extremely painful upon palpation, knee flexion was lost and he was unable to stand, so magnetic resonance imaging (MRI) was performed. MRI showed osteomyelitis of metaphysis and distal diaphysis of the right femur with associated subperiosteal purulent collection and concomitant pyomyositis and fasciitis of the distal right thigh. Blood cultures were positive for methicillin-resistant *Staphylococcus aureus* (MRSA). The patient received a long course of intravenous antibiotic therapy and his condition slowly improved. Follow-up femur X-ray showed a mixed pattern of erosion and sclerosis at the meta-diaphyseal region and periosteal reaction at the diaphyseal region. This case highlights the importance of early scabies diagnosis even in Western countries where poverty and household overcrowding are uncommon. Early diagnosis, timely initiation of proper treatment and evidence of clinical resolution are important elements to prevent recurrence of infection and serious systemic superinfections even from multi-drug resistant bacteria. Clinical consequences from unrecognized disease or inadequate eradication are preventable.

## 1. Introduction

Scabies is a skin infestation from the mite *Sarcoptes scabiei*. Interpersonal transmission is primarily caused by prolonged skin-to-skin contact or, less frequently, may occur via clothing, bedding or towels [1]. Scabies is endemic in developing countries and is considered a public health issue of great concern. It is one of the neglected tropical diseases listed by the WHO [2]. In high-income settings, the disease occurs sporadically, but outbreaks may occur [3]. Scabies infestation can occur at any age, but most cases occur in pediatric patients [4,5], with the highest prevalence in infants younger than 2 years old [6]. Classic scabies generally presents 4 to 6 weeks after infestation [7]. Skin lesions consist of multiple papules, which can be localized in every area of the skin, although fingers, wrists, elbows, axillary folds, genitalia and extensor surface of the knees are the areas most involved [8], probably due to lipid composition and other site-specific features of the skin in these body regions [9]. Sometimes burrows, produced by female mites after mating, can be noted on the skin. The most relevant clinical feature is pruritus, which can be severe and worsens at night [10] and can be relieved by the administration of oral antihistamines. In infants, itching may cause irritability and refusal to eat [4]. Crusted scabies, also known as Norwegian scabies, is a clinical variant of scabies that is highly contagious as it is associated with a heavy mite burden [11]. Crusted scabies is rare in children and usually affects immunocompromised hosts [5]. Skin usually appears “crusted” with widespread hyperkeratotic papules and fissured plaques [12]. 

The diagnosis of scabies is generally made on the basis of skin findings and reported symptoms, especially itching, rather than identifying mites from skin scratches [10]. Dermoscopy is a useful non-invasive diagnostic tool that can help visualize *Sarcoptes scabiei* on the affected skin [13]. Management of scabies needs effective therapies and identification and treatment of members of the family and other close contacts [14]. Etiological treatment is based on anti-scabietic agents such as topical permethrin and oral ivermectin [8,15]. The latter should preferably be administered in children weighing more than 15 kg but recent evidence suggests its use is safe even in infants [16]. Without proper treatment, scabies could cause scratch lesions, which may be the entry point for opportunistic and pathogenic bacteria [17]. Bacterial infection of the skin, called impetigo, and soft skin infections are quite common during scabies infestation [10,18] and usually due to *Staphylococcus aureus* and *Streptococcus pyogenes* [19]. The same bacteria are also involved in less frequent systemic infections, such as osteomyelitis, which are a matter of concern [20]. 

## 2. Case Report

We report the case of D.A. a 3-year-old boy, born in Peru who moved to Italy with his family when he was 18 months old. He had no relevant past medical history, apart from febrile seizures at 9 months old. 

After moving to Italy, D.A. and his family were hosted by friends in a small house in the Milan suburbs. During their stay, D.A. and one of the hosts were diagnosed with scabies and treated with local application of permethrin. Unfortunately, D.A. relapsed a couple of months later, but this time his parents gave him no treatment. 

About 1 month after the scabies recurrence, he was taken to the emergency department because of a 4-day history of fever, right thigh pain and refusal to walk. He had severe and diffuse itching for many weeks. 

### 2.1. Diagnosis

Upon medical examination he looked generally unwell, pale-looking and a bit irritable. His right thigh was extremely painful on palpation, knee flexion was lost, while no limitations in hip movements were evident. He was not able to stand or walk. No local signs of impetiginization were found, but multiple crusted-papular lesions were present on the lower limbs and abdomen, as well as many scratch lesions. 

Blood tests showed neutrophilic leukocytosis (white blood cells 14,600/mm^3^ with absolute neutrophil count 10,600/mm^3^), significantly increased C reactive protein (CRP) (118 mg/L, nv < 10 mg/L), normal procalcitonin (0.57 µg/L, nv < 0.5 µg/L) and mildly elevated liver enzymes (ALT 110 U/L, AST 96 U/L). Fibrinogen was also elevated (917 mg/dL), as well as D-dimer (973 ng/mL). Right hipbone and femur X-rays showed no bone fractures, hip and knee ultrasounds were negative. A contrast enhancement MRI of the pelvis and bilateral thighs, under-sedation, was performed. MRI demonstrated abnormal bone marrow signal intensity, consistent with osteomyelitis, in the metaphysis and distal diaphysis of the right femur, which appeared inhomogeneous in relation to the presence of edematous regions and hypo/avascular areas. The periosteum was elevated due to an associated subperiosteal purulent collection and there was concomitant pyomyositis and fasciitis of the distal right thigh and an abscess within the obturator externus muscle of the contralateral limb (Figure 1A,B). The diagnosis of acute bacterial osteomyelitis of the right femur with concurrent cellulitis, pyomyositis and fasciitis was given. Skin lesions suggested a recurrence of scabies and the diagnosis was confirmed by a dermatologist since many scabies mites in burrows were found at dermoscopic evaluation.

### 2.2. Treatment Plan

D.A. was admitted to our Pediatric Department. After obtaining blood cultures, an empirical combination of intravenous vancomycin and cephazolin was administered. Forty-eight hours after starting the antibiotics D.A. was still febrile and his general conditions had worsened. Based on muscular and soft tissue involvement in MRI images, intravenous metronidazole was added to the ongoing antibiotic therapy to ensure better coverage for anaerobic bacteria. The trans-thoracic echocardiogram was negative for valvular vegetations. Blood tests showed an increase in CRP (346.5 mg/L) and procalcitonin (6.5 µg/L); blood cultures were positive for methicillin-resistant, vancomycin-susceptible *Staphylococcus aureus*. Since scabies diagnosis was confirmed, another course of local therapy with permethrin was administered and ivermectin was also prescribed to all involved households.

### 2.3. Treatment Progress

By day 5 of the antibiotic course, D.A. became afebrile and his general conditions improved significantly. In days that followed, thigh pain was also resolved and no more limitations or pain at knee flexion were noted. CRP became negative on day 10 of antibiotic therapy, as well as liver enzymes, fibrinogen and D-dimer. Metronidazole was stopped after 10 days, while cephazolin was continued for 4 weeks. An 8-week course of intravenous vancomycin was completed. X-rays of femur, performed during week 3 of hospitalization, showed a mixed pattern of erosion and sclerosis at the meta-diaphyseal region and there was a periosteal reaction at the diaphyseal region. D.A. started walking after 4 weeks of hospitalization. Despite a negative personal medical history, we performed a first-line immunologic screening (Ig levels and lymphocyte phenotyping) which resulted as being normal. Scabies was also resolved and we did not observe any recurrence during hospitalization.

## 3. Discussion

In subtropical and tropical areas, the prevalence of skin infections in patients affected by scabies is high [21], although data are controversial [18]. Depending on the geographical area considered, *S. aureus* is one of the most relevant bacteria that causes infections, together with group A beta-hemolytic *Streptococci* [19]. In Western countries, skin bacterial infections in immunocompetent adult patients affected by scabies are rare, but in pediatric population impetigo is quite common and *S. aureus* is involved in almost all cases [22].

Secondary skin and soft tissue bacterial infections by *Streptococcus* and *Staphylococcus* are well documented [23], and some cases of systemic bacterial infections are also reported in the literature, even if they are mostly as a result of a complication of the crusted variant of scabies. Invasive infections from *S. aureus* are linked to a 5% case fatality rate [20]. Secondary infections by group A *Streptococci* may result in symptomatic or pauci-symptomatic acute glomerulonephritis [1].

Mallo-Garcìa et al. [24] described the case of a 2-month-old infant who developed an abscess of the knee just below scabies lesions. The soft tissue infection was the portal of entry for *Staphylococcus aureus*, which caused sepsis with severe respiratory distress due to empyema and hemothorax. The patient had clinical improvement after an intravenous course of cloxacillin. Osteomyelitis as a complication of scabies is rare. As far as we know, only one case of osteomyelitis after impetiginized scabies is described in the literature [25]. A 15-year-old boy with a 3-month history of scabies unresponsive to treatment, was admitted to the pediatric department seriously ill, febrile and with severe lower limb pain. His blood cultures and skin swabs were positive for *S. aureus*. Imaging findings showed distal femoral osteomyelitis and a 5-week course of intravenous oxacillin was performed with a good clinical response.

### Strenghts and Limitations

A limitation of our study is that we did not perform skin swabs to our patient who presented with multiple crusted-papular lesions, without any evidence of acute impetiginization. However, considering that *Staphylococcus aureus* is a commensal bacteria of the human skin flora and most frequently involved in skin infections [26], we can assume that the untreated cutaneous lesions due to *Sarcoptes scabiei* were the portal of entry for *S. aureus*, leading to hematogenous spread of the bacteria and then to fasciitis, pyomyositis and acute severe osteomyelitis. 

As clinical consequences, including severe complications, from unrecognized scabies or inadequate eradication are preventable, the strength of our case is to highlight the importance of making an early diagnosis, administering prompt and appropriate treatment and confirming the clinical resolution of scabies. If symptoms fail to improve after 1–2 weeks, clinicians should consider poor adherence to treatment, resistance to therapy or a possible incorrect initial diagnosis. A microscopic examination may be performed at 2 weeks after completion of treatment to confirm eradication [27,28]. Recurrence of the disease should be considered too. Some studies of the adult population have reported that scabies recurrence is not uncommon and may result from treatment failure or reinfection from untreated contacts [29,30,31,32]. For this reason, cohabitants and all individuals with prolonged physical contact should also be timely and correctly treated to reduce the risk of reinfection [33].

## 4. Conclusions

In conclusion, even in those regions where the disease is not endemic, in cases of suspicious skin lesions, infection by *Sarcoptes scabiei* should always be taken into consideration by evaluating the presence of risk factors, such as crowded households. Scratch lesions determined by scabies, acting as portals of entry, may cause systemic bacterial spread with secondary localization. Osteomyelitis may be a life-threatening infection, especially when sustained by drug-resistant pathogens, as in our case, and late diagnosis may be associated with a long healing process and severe bone disease, which may eventually affect growth and progress to osteonecrosis, chronic osteomyelitis and bone destruction.

## Figures and Tables

**Figure 1 ijerph-19-01007-f001:**
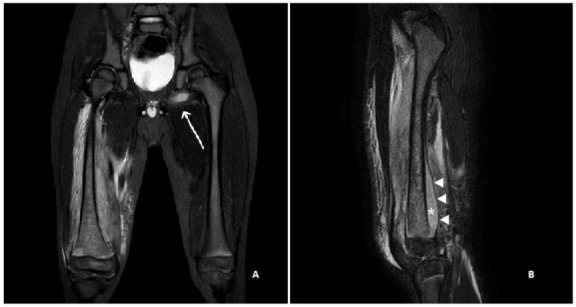
Coronal (**A**) and sagittal (**B**) STIR (short tau inversion recovery) MR images show abnormal bone marrow oedema in the distal femur, with marked irregularity of the signal intensity and an associated large subperiosteal abscess (*) due to accumulation of pus beneath the elevated periosteum (arrowheads). Fasciitis and pyomyositis of the right thigh as well as an abscess in the contralateral obturator externus muscle (arrow) are also evident.

## Data Availability

Not applicable.

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
