# Peer review of "MRSA Femoral Osteomyelitis from Superinfected Scabies Lesions: A Pediatric Case Report"

_ijerph, 2022, doi:10.3390/ijerph19021007_

Round 1

Reviewer 1 Report

Peer-Review Report

Dear Authors,

Manuscript ID ijerph-1425911

Manuscript Title: MRSA femoral osteomyelitis from superinfected scabies lesions: A pediatric case report

            Congratulations, you presented a novel and an interesting case of MRSA-induced osteomyelitis from a superinfected scabies in a three-year-old male child; the case, I believe, possesses high interest for readers from two disciplines of medicine, including dermatology and pediatrics. I have recommended the managing editor of the IJERPH accepting your case report article pending "minor revisions"; these include corrections to minor methodological errors, text editing, and a moderate level of English language proofreading. 

            Further, there are two significant defects in your writing;  the primary defect within the full-text article is that you did not correctly follow the stylistic writing requirements of the IJERPH, especially concerning the in-text and bibliographic citations; some sentences do not have references. The second important thing, an ethical concern, is that you should provide the journal's editor with a copy of the "written consent" that you acquired, from the parents, concerning this case report.

            I am attaching the full-text article (in PDF) with highlights and comments to complement this peer-review report; you should abide by and correct your manuscript according to each.

  • The title is excellent and representative of the dimensions of the case report.
  • For a case report, there is an excessive number of authors (six of you).
  • The abstract is good; however, the author must mention the local clinical findings that led them to implement MRI.
  • The authors should select and expand on the keywords by implementing Medical Subject Headings (MeSH); available from the National Library of Medicine (NLM), at https://www.ncbi.nlm.nih.gov/mesh/
  • The introduction section is brief and needs to be further expanded. The authors used only two reference materials for the introduction section, which is inadequate. Besides, the authors did not adhere to the stylistic writings of the guidelines of the IJERPH.
  • Line-37. The authors should revise the in-text citations as per the IJERPH requirements and instruction for authors, available from https://www.mdpi.com/journal/ijerph/instructions. Besides, all statements and sentences in the introduction and discussion sections require proper referencing. I have noticed several statements that do not have in-text citations of reference material.
  • Line-40 and 44. Where are the references? Kindly check the full-text manuscript for sentences without referencing.
  • The case report section is elegant and describes the holistic case scenario, diagnosis, and management; it represents the best section of the article.
  • Line-77. The MRI imaging is vital for the case report; the (para)sagittal MRI view provided interesting radiographic findings
  • The discussion section is good and adequate.
  • The conclusion section is good and highlights important clinical recommendations.
  • Line-157. The authors should either delete the "Patent" section or write "Not Applicable".
  • Line-165. The authors should provide the written consent to the journal's editor; this is mandatory.
  • Line-172. The authors should format all of the in-text citations and bibliographic citations in compliance with the IJERPH instructions for authors; available from https://www.mdpi.com/journal/ijerph/instructions#references

All the best,

The peer-reviewer.

Author Response

Dear Reviewer,

thank you very much for your precious suggestions. We have modified our paper accordingly and this will surely improve the quality of the manuscript.

English language has been revised by a native English speaker to improve the grammar and readability.

Point by point answers:

  • We have modified the papers in order to adhere to the stylistic writing requirements of the IJERPH, as suggested.
  • Informed consent has already been sent to the managing editor, according to the instructions received, at the very early stages of submission and it was accepted.
  • New keywords have been added following Medical Subject Headings (MeSH) terms, as suggested.
  • We have expanded the introduction section and added references, as suggested.

Line-37: we have revised the in-text citations as per the IJERPH requirements and we have provided proper referencing for statements and sentences of the section.

Line-40 and 44: we are now providing referencing for statements and sentences of the section.

Line 157: correction done.

Line-165: we have already solved the issue about parents’ consent with the managing editor, see above.

Line-172: references have been formatted in compliance with the IJERPH instructions for authors

Reviewer 2 Report

The presented article in the form of case study is not at all complete, I see a lots of sentences finishing with no any valuable convey, references was cited unproper, the Introduction part scaly presented based on 13 literatures. 

The following:

This case highlights the importance of early diagnosis, proper treatment and evidence of clinical resolution of scabies. In fact, reports from worldwide countries point out that scabies recurrence, in adult population, is not uncommon and may result from treatment failure or reinfection from untreated contacts 9−11. 

Very common information, nothing as a discussion or steady conclusions. 

Treatment failure may be due to poor adherence or resistance to therapy. Eradication is likely if scabies lesions resolve by 142 one week after treatment12.

The authors cannot expressed a strong hypothesis about the case. 

Author Response

Dear Reviewer,

thank you very much for your precious suggestions. We have modified our paper accordingly and this will surely improve the quality of the manuscript.

English language has been revised by a native English speaker to improve the grammar and readability.

Point by point answers:

  • We have revised the manuscript; in particular, we have improved the introduction section, that you have specifically described as weak and incomplete, with new information and addition of references, as suggested
  • We have modified the last sentences of the discussion paragraph, as suggested
  • “Treatment failure may be due to poor adherence or resistance to therapy. Eradication is likely if scabies lesions resolve by one week after treatment. The authors cannot expressed a strong hypothesis about the case”: we have modified as suggested.

Round 2

Reviewer 2 Report

The level of the present called by the authors "revised'' form is still far from the article. This was a small informative note.

Author Response

Dear reviewer, 

thank you very much for going through revision of our paper again.  We have provided a certificate of the language check made by an English native-speaker.  As for your comment, we kindly ask you to provide us with some more details and suggestions.  Thank you. 
